# Performance Properties of Cement–Glass Composite Bricks (CGCB) with Additively Manufactured (AM) Polymeric Scaffolding

**DOI:** 10.3390/ma16051909

**Published:** 2023-02-25

**Authors:** Marcin Małek, Janusz Kluczyński, Waldemar Łasica, Mateusz Jackowski, Ireneusz Szachogłuchowicz, Jakub Łuszczek, Janusz Torzewski, Krzysztof Grzelak

**Affiliations:** 1Institute of Civil Engineering, Faculty of Civil Engineering and Geodesy, Military University of Technology, 2 Gen. S. Kaliskiego St., 00-908 Warsaw, Poland; 2Institute of Robots & Machine Design, Faculty of Mechanical Engineering, Military University of Technology, 2 Gen. S. Kaliskiego St., 00-908 Warsaw, Poland

**Keywords:** additive manufacturing, cement–glass composite bricks, digital image correlation analysis, material extrusion, fused filament fabrication, PET-G, waste disposal

## Abstract

This study provides an alternative to traditional masonry materials: a cement–glass composite brick (CGCB), with a printed polyethylene terephthalate glycol (PET-G) internal scaffolding (gyroidal structure). This newly designed building material consists of 86% waste (78% glass waste, and 8% recycled PET-G). It can respond to the construction market’s needs and provide a cheaper alternative to traditional materials. Performed tests showed an improvement in thermal properties after the use of an internal grate in the brick matrix, i.e., an increase in thermal conductivity (5%), and a decrease in thermal diffusivity (8%) and specific heat (10%). The obtained anisotropy of the CGCB’s mechanical properties was much lower than the non-scaffolded parts, indicating a very positive effect of using this type of scaffolding in CGCB bricks.

## 1. Introduction

Concrete, as a material, is characterized by high, or very high, compressive strength during static testing. The high compressive strength, significantly increases the brittleness of concrete, with low strain [1]. Conventional steel reinforcement, used in building engineering, is the most popular solution for low material properties during some specified load conditions (i.e., tensile strength at bending). The dramatic growth in the steel price, caused by the COVID-19 pandemic and the war crisis in Ukraine, at the beginning of 2022, has encouraged the development of novel types of concrete–polymer composites [2].

This approach is attractive, since it is possible to additively manufacture the internal reinforcement (called scaffolding). Salazar et al. [3] suggested using a polymeric scaffolding for constructions exposed to four-point bending: cheap, ultra-high-performance concrete (UHPC). The authors [3] considered two types of polymeric reinforcement structures, three-dimensional octets and cubic lattices. A conventional form of concrete reinforcement, using bars, cannot be used in a fully controlled manner, increasing the value of the stochastic factor of placement and orientation of the reinforcing bars. The new concept of design and topological optimization of the internal scaffold for concrete bricks, involves changes to the amount of the material in stress concentration areas [4,5,6,7,8]. The main conclusion of the Salazar et al. [3] work, was the lack of a connection between the material type and the final strength of the concrete–plastic composite. The primary influence on the sample strength was the scaffolding type, not the material type. A different solution to the reinforcement of concrete composites was suggested by Xu et al. [9]. The authors used the finite element method (FEM) to properly prepare a polymeric scaffolding for exact usage in the construction. The main aim of their work was to increase the total strain of the tested samples during three-point bending. The geometry of the samples had been prepared based on the maximum stresses generated during preliminary bending tests. Such a prepared research methodology allowed the elimination of the brittle-like cracking during the sample’s fracture. Additionally, a specially designed scaffolding shape and volume, significantly affected the final properties of the manufactured concrete–polymer composites. Another solution, the cement bricks composite (CBC), with the use of an AM polymeric scaffold, was suggested by Qin et al. [10]. The authors used varying geometries of the structure, based on a combination of basic shapes such as hexagons, squares, diamonds, etc. The main aim of their work was similar to [9], i.e., increasing the total strain of the material during bending testing. The authors [10] also used different materials for the AM of the scaffolding: polyamide (PA) filaments and polymeric resins. The final results indicated that PA structures are a much better solution than reinforcement obtained by the AM of resins, especially transparent resins. A quite different approach to using AM in building engineering was shown in Fadeel et al. [11], where the authors produced a new type of form dedicated to brick manufacturing. The mentioned form allowed the production of parts characterized by very complex structures, to increase their total compressive strength. This approach was made possible by using finite element method (FEM) analysis to optimize the final scaffolding geometry, made of ABS material, from the energy dissipation point of view. The CBCB is not the only field of research currently using AM technologies, concrete mortars are also in the mainstream of scientific research. Quite a new approach was shown by Salazar et al. [12], in their work on using an AM spatial mesh made of recycled thermoplastic polymers. This kind of reinforcement provided a two-fold increase in the four-point bending strength. An additional positive side effect of using this kind of reinforcement, is that it ensured better workability of the produced concrete mortar, compared to modified solutions using reinforcing fibers. A different method of using AM in concrete mortars has been demonstrated by Lin et al. [13]. Compared to other technologies, the authors used recycled ABS filaments for the AM processes of reinforcing flat structures. Such structures were deposited with the concrete layer-by-layer, positively affecting the strain–stress characteristics.

In all of the abovementioned technologies using AM in building engineering, researchers have mainly used polymeric scaffolding, with conventional concrete. To increase the number of waste materials in the construction parts, additional elements of the concrete mortar could be used, e.g., waste glass as an aggregate material [14]. The use of PET-G for additional reinforcement is justified by its high chemical resistance [15,16,17] and good mechanical properties [18,19]. Due to the difference in the materials used for the production of the concrete composite, i.e., the ceramic material and the polymer, the authors noticed that there is a physically justified difference between the thermal conductivities of both materials. As a result of their physical connection, there are no reaction processes between the concrete and the polymer, therefore chemical adhesion does not occur in this case. According to the literature, in such a case, the combination of polymers and ceramic materials leads to the so-called mechanical adhesion, i.e., due to the geometry of the 3D print, the structure was “anchored” in the designed concrete. In such a case, there is a need to check whether, apart from the increase in mechanical parameters, there is no decrease in other parameters that are determinants for concrete in construction, i.e., its insulation. According to the results of the obtained tests, the insulating power of such a composite increased, which reassured the authors of the validity of their work. The increase in insulation was achieved by creating an air layer, which is an ideal insulator, between the contact boundaries of both materials. Despite the significant growth in pro-ecological movements in building engineering, many issues have to be addressed from the point of view of the material’s behavior. An additional factor, in the case of this research, is the use of polymeric scaffolding, which needs to be motivated. In the case of a previous study [14], there was a positive effect with the use of scaffolding, but the reinforcing structure did not properly penetrate the scaffold. Therefore, in this study, AM polymeric scaffolding was used to produce a CGCB brick, with waste glass as an aggregate material. The geometry of the scaffold was modified to allow proper penetration of the waste glass parts. In the available research, there is not enough data about using a mixture of waste polymers (as scaffolding) and waste glass (as reinforcing structures). Hence, mechanical and thermal analyses of such materials were made, to determine the basic performance properties of such kinds of composite materials. The suggested approach could be helpful in a further analysis of using waste materials in building engineering.

## 2. Materials and Methods

### 2.1. Characteristics of the AM Process and Polymeric Waste Material

Material extrusion (ME) technology was used to produce the internal polymeric scaffolding. The manufacturing process was held on the fused filament fabrication (FFF) device, Prusa Original MK3s (Prusa Research, Prague, the Czech Republic), with the use of PET-G materials, in the form of a 1.75 mm filaments (Spectrum Filaments Ltd., Pecice, Poland). Before the process, the material was dried in the laboratory dryer for 4 h at 65 °C (using the manufacturer’s instructions). The processing parameters used were as follows:Hotend temperature: 240 °C;Substrate plate temperature: 85 °C;Layer thickness: 0.2 mm;Infill: 100%;Part cooling intensity: 40%;Printing speed: 50 mm/s;Nozzle diameter: 0.4 mm.

As a geometrical scaffolding structure, a gyroidal infill structure was chosen, to ensure an easy flow of the waste glass parts. Such a geometry is available as a default shape of the infill in the PrusaSlicer Software (version 2.4.1), dedicated to the FFF process preparation. To generate a structure as a form of infill, a full monolithic 40 × 40 × 160 mm^3^ block was used. In the further steps, the infill amount was set to 5%, and external shells (outline, bottom, and top) were completely removed. The final form of the reinforcing structure is shown in Figure 1.

### 2.2. Characterization of the Filler: Cement–Glass Mortar

Based on Portland cement (CEM I 42.5R NA [20]), whose specifications were established following EN 196-6:2019-01 [21] and PN EN 196-1:2016-07 [22], tap water, waste glass powder (grain size less than 0.1 mm), and waste glass aggregate (grain size less than 2.0 mm), the cement–glass mortar was proposed as a filler for the CGCB. The composition of the final component’s grain sizes (entirely extracted from glass waste packages) was, about 2%, 8%, 14%, 25%, 21%, and 30%, for 0–0.063 mm, 0.063–0.125 mm, 0.125–0.250 mm, 0.250–0.500 mm, 0.500–1.000 mm, and 1.000–2.000 mm, respectively. A well-compacted aggregate pile was achieved by the glass–sand–grain composition, fitted between upper and lower guidelines [23,24]. The glass cullets employed in this investigation had an uneven surface form, which was created by a mechanical or implosive crushing procedure. Additionally, green, brown, and transparent glass particles were employed in this study. Furthermore, a liquid third-generation additive, based on modified polymers, was used to maintain a low water-to-cement ratio (w/c = 0.29), in order to drastically minimize the amount of tap water in the cement–glass mortar filler. Table 1, Table 2 and Table 3 provide information on the cement and glass waste’s chemical make-up and physical characteristics, respectively.

### 2.3. The Manufacturing Process of Cement–Glass Composite Brick

The formula for the cement–glass mortar was produced using generic, already-in-use techniques for developing high-quality composites [26]. The approach of computational–experimental design was employed. Currently, there are no computational approaches that might guarantee the potential of making high-strength composites with repeatable outcomes without actual experiments, which is why manual stress tests were carried out. General calculations and assumptions were established in the design process, which were later refined after being empirically confirmed (during laboratory tests). The final formula of the CGM [14], used as a filler to create the concrete brick with 3D printed scaffoldings, was, 480 kg of cement, 140 kg of water, 4.8 kg of chemical admixture, 117.8 kg of glass powder, and 1782.2 kg of glass aggregate.

A high-speed planetary mixer, with three ranges of stirrer rotation speed, was used to combine all the dry ingredients mentioned above, for one minute. After adding the wet components, the mixing procedure lasted for an additional four minutes. A medium speed was selected, to thoroughly combine all ingredients (second of three possible rates). The CGM was then compressed on a vibrating table, after being put into molds with previously printed scaffolding. The vibration duration for a single layer was about 30 s. Then, using a regular knife, dampened with water, the top layer of the sample was leveled with the edge of the mold, after the mold had been filled. To prevent excessive evaporation of the mixing water, and shrinkage strains brought on by the heat of the cement hydration process, the upper layer of the sample was covered with absorbent mats twenty-four hours after it was manufactured. A 12 h pre-treatment period was involved. The samples were then de-molded and kept in water, by EN 12390-2:2019-07 [27]. The laboratory conditions during the whole manufacturing process were, 21 °C temperature and 50% humidity.

### 2.4. Physical Properties Testing Methodology of Cement–Glass Composite Brick

To pinpoint the thermal characteristics of the hardened CGM, the thermal conductivity, thermal diffusivity, and specific heat of the hardened samples were studied. The ISOMET 2114 analyzer was used for all measurements (Applied Precision Ltd., Bratislava, Slovakia). The resistor heater on the analyzer’s probe was close to the sample being tested, making it possible to measure how the material’s temperature responded to heat flow pulses. Also employed was an analyzer, with a 60 mm diameter probe. Ten specimens, in the form of cubes, with dimensions of 150 mm × 150 mm × 150 mm, were evaluated, to determine all of the CGM’s thermal characteristics. These samples were also used to test the density of the hardened samples. This was computed by dividing the mass by the sample volume. Also, the same procedure was performed on ten final CGCB specimens to compare the results, and indicate the influence of the interior PET-G scaffolding on the brick’s properties.

### 2.5. Mechanical Properties Testing Methodology with a Digital Image Correlation (DIC)

The compression and bending strength tests for the CGCB and CGCB with polymeric scaffolding samples were performed using the Instron 8802 (Instron, Norwood, MA, USA) testing machine, shown in Figure 2. Bending tests were performed on a test rig, with a support distance of 120 mm. Both bending and compression tests were on the same samples (40 mm × 40 mm × 160 mm). All tests were conducted employing DIC measurements, utilizing a Dantec Dynamic (Dantec, Ulm, Germany) device and the ISTRA 4D software.

## 3. Results and Discussion

### 3.1. Physical Properties of Concrete–Glass Composite Brick

#### 3.1.1. Density

The CGCB samples acquired a reduced density equivalent to 2051 kg/m^3^, but the CGM sample had an average density of 2157 kg/m^3^ [14]. Both concretes were assigned to the standard concrete class [28]. Using additional polymeric scaffolding caused a density reduction of about 5% (a difference of 106 kg/m^3^). Such a phenomenon was caused by the significant mass difference between the cement–glass mortar and the PET-G material. Additionally, the tension between the plastic scaffolding and glass cullets indicated the presence of more air voids, that lowered the brick’s density. Małek et al. [14] presented similar trends when using the three dimensional structure “3Dhon” for scaffolding. In their study, the final CGCB reached a density of 1982 kg/m^3^, and was classified as a lightweight concrete, with D2.0 class [28]. This disparity results from the design of the interior scaffolding, which was thicker and took a greater mold volume than the gyroidal structure tested in this study. Despite this difference, it confirms a downward trend in terms of the density of the cement–glass mortar with the printed PET-G reinforcement. Also, the CGCB showed about the same density as traditional bricks, tested by Bautista-Marín et al. [29], without any additions (about 2200 kg/m^3^).

#### 3.1.2. Thermal Tests on CGM and CGCB

The hardened CGCB was tested for its thermal properties. The average values of thermal conductivity, thermal diffusivity, and specific heat were calculated based on ten measurements. As presented in Table 4, compared to the cement–glass mortar [14], the final CGCB showed about a 12% decrease in thermal conductivity (0.87 ± 0.05 W/mK), and about a 20% decrease in specific heat (1.31 ± 0.01 MJ/m^3^K). Only thermal diffusivity increased after incorporating the PET-G scaffolding with the gyroidal structure into the matrix. The increase was from 0.61 ± 0.03 µm^2^/s to 0.69 ± 0.03 µm^2^/s. Based on available research [14], the gyroidal PET-G scaffolding showed similar trends to the 3Dhon scaffolding incorporated into a cement–glass mortar matrix. However, samples tested by Małek et al. [14], showed lower changes to the thermal properties: 9%, 10%, and 8% for thermal conductivity, specific heat, and thermal conductivity, respectively.

### 3.2. Compressive and Bending Tests with the Use of DIC

#### 3.2.1. Three-Point Bending Tests

The flexural strength was determined for samples oriented in the vertical and horizontal directions (shown in Figure 1). To better understand the influence of the scaffolding used, the results of the test (shown in Figure 3) include concrete samples without scaffolding (CGC vertical and horizontal), separated polymeric reinforcement (reinforcement vertical and horizontal), and the manufactured composite of concrete and polymeric scaffold (composite vertical and horizontal).

The measured flexural strength of CGC, oriented vertically and horizontally, reached values of 2.4 kN and 2.1 kN, respectively. It is seen that, in the vertical direction, samples are characterized by an almost two times greater beam deflection (from 0.22 mm to 0.4 mm). The use of the additional polymeric scaffolding inverted this phenomenon. It caused the properties to be more isotropic from the beam deflection point of view (the deviation of the representative curves is visibly smaller). Another important advantage of using polymeric reinforcement with a gyroidal geometry, is that it caused a 5% increase in the total force, in the case of both (horizontal and vertical) directions. Another significant statement could be made based on the DIC results (Figure 4). There is not any significant difference between the CGCB with or without polymeric scaffolding. This phenomenon allows us to state that, from a mechanical properties point of view, the material’s behavior during static loading is similar to conventionally made bricks, without additional scaffolding. Such positive results could be justified because of the high level of the mortar’s distribution between the polymeric structure cells. The glass particles went through the bigger cells in the designed scaffold, which positively affected the fracture behavior stability in both orientations for the tested samples. The visible crack course in Figure 4a can be compared to Figure 4c, and that in Figure 4b to Figure 4d, respectively. Such a phenomenon proves the stability of the material during static loading.

CGCB showed greater than 5 MPa flexural strength, usually taken as a standard for masonry materials [30]. Its flexural strength of 5.90 MPa in the horizontal orientation, and 6.75 MPa in the vertical orientation, proves that it can be a replacement for traditional masonry materials as a novel eco-building material.

The obtained DIC results do not differ significantly from the data available in the literature. Shah et al. [31] analyzed mortar bricks using the DIC method during three-point bending tests. The authors registered a single crack tip along the sample part, with significant amounts of microcracks in the fracture zones that form ahead of the traction-free crack, a property of a quasi-brittle material. A visible increase in total deformation of horizontally oriented composite samples (Figure 3), decreases the typical behavior of cracking (which is proved by a more complex crack tip in the case of the horizontally oriented samples). Quite a different approach, more suitable for CGCB samples with AM scaffolding, was suggested by Wu et al. [32], where they used a cohesive crack concept, that allows a description of the strain-softening behavior of the material. This approach finds the relationship between the fracture process zone and the crack length. The results obtained utilizing the DIC method help to better study the behavior of the crack extension resistance curve of the material. Based on this method, and the DIC measurements, it could be observed that the length increased during crack propagation, but decreased after the fracture process zone fully developed at a crack extension.

#### 3.2.2. Compressive Tests Results and Discussion

A positive effect of using internal polymeric scaffolding is also visible in the case of the compressive testing. The representative courses of each combination are shown in Figure 5. The reinforced CGCB is characterized by more isotropic properties than registered in the samples without any scaffolding. The courses of the curves for the horizontal CGCB samples, with and without scaffolding, are similar, and the same phenomenon is visible in the vertical direction. In the case of the vertical direction, the courses of the curves are almost the same. This shows that this type of internal scaffolding has much better properties than those registered in the case of our own previous research [14].

Additionally, the designed cement–glass composite brick’s compressive strength is in line with traditional bricks, tested by Zuo et al. [33]. They showed about 24 MPa compressive strength, compared to 25.6 MPa for CGCB in the horizontal orientation. The value reported for vertical orientation was much lower—14.4 MPa—because of the asymmetry of the sample. The glass grain sizes varied between horizontal and vertical orientations due to their random distribution.

The main difference was related to the polymeric structure’s internal cell dimensions, and the gyroidal structure used in this research allowed for a significantly better glass particle distribution. Such an approach obtains the essential advantage of composite structures—a combination of the properties characterizing both materials. The described phenomenon is depicted in Figure 6, in the microscopic fracture images. Each cell has been evenly filled with glass particles.

Another analysis connected with the DIC measurements, revealed a positive influence of the scaffolding structure used, partially shown in the compressive testing curves. In the case of the reinforced samples, the increased strain areas are significantly larger than in samples without additional scaffolding. This kind of phenomenon is visible in both the vertical (Figure 7i) and horizontal (Figure 7l) orientations. This answers why additional scaffolding changed the material’s characteristics to being more isotropic.

A practical method, that could be used for a proper description of this kind of material’s behavior, was described by Zhou et al. [34], for rock-like specimens. The stress–strain curves in this methodology were divided into two stages:-the strain-softening stage;-the residual strength stage.

The strain-softening stage is characterized by rapid crack evolution and further propagation, leading to a rapid drop in the axial stress. With the increase in uniaxial compressive loads, the axial stresses decrease to the residual strengths in the materials with different brittleness indices (i.e., polymeric scaffold or waste glass reinforcement). Such an approach would be possible by analyzing particular cells of the scaffold.

## 4. Conclusions

The main aim of this research was to determine the mechanical and thermal behaviors of the proposed CGCB bricks. An additional purpose was to compare scaffolded and non-scaffolded test samples. Such an approach allowed for a deep analysis of the use of additional polymeric structures. The results revealed the positive physical properties (thermal and mechanical) of a new type of concrete with glass particles, additionally reinforced by an additively manufactured polymeric structure. The analyses conducted highlight the positive influence of using waste materials, such as glass particles or recycled PET-G. The combination of these two materials in building engineering solutions is also possible. Based on the mixtures law, it could be additionally concluded that the addition of such kinds of waste materials increases the chemical resistance of the produced bricks. The results of this research allow us to draw the following conclusions:The cement–glass composite brick, with a printed PET-G gyroidal structure, has better thermal properties than the cement–glass mortar itself, as thermal conductivity after the modification decreased by 12%, and specific heat was reduced by 20%. The thermal diffusivity, on the other hand, increased by 13%.After incorporating the gyroidal structure from PET-G into the cement–glass mortar matrix, the density changed from 2157 kg/m^3^ to 2051 kg/m^3^.The use of the AM gyroidal structure made the concrete brick’s mechanical properties more isotropic, with up to 50% lower beam deflection during the three-point bending test.

To obtain more stable properties of the material, in different directions, it is necessary to use polymeric structures with proper cell dimensions—it is essential to allow the easy flow of the glass particles in the material’s volume.

For further research, it is necessary to create a structure that could improve the mechanical properties of the produced bricks. Such a positive effect in this research (as more isotropic properties), could be improved by increasing the volume of the scaffolding material.

The novel cement-glass composite brick, made from approximately 86% of waste, is an example of the reuse of rubbish from other industries in the construction sector. It should be noted that the durability of the final CGCB is comparable to traditional bricks, while ensuring increased thermal insulation at the same time.

## Figures and Tables

**Figure 1 materials-16-01909-f001:**
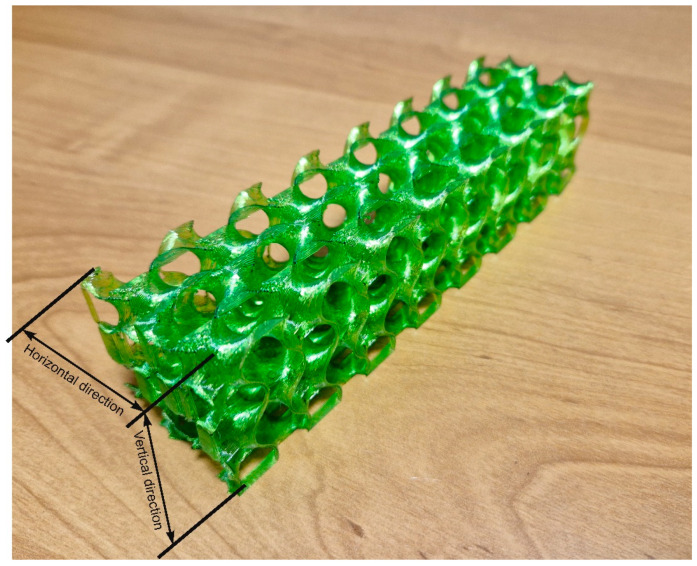
Gyroidal reinforcement structure.

**Figure 2 materials-16-01909-f002:**
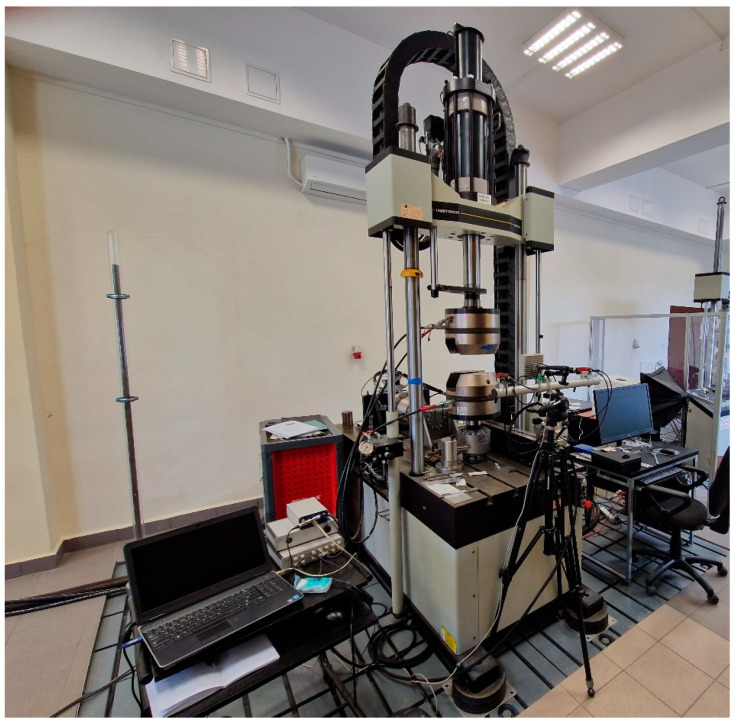
Instron 8802 servo-hydraulic pulsator for three-point bending and compression tests, equipped with the Dantec DIC system.

**Figure 3 materials-16-01909-f003:**
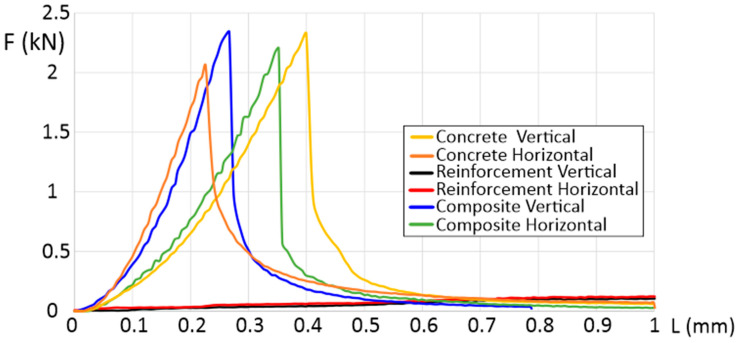
Force as a function of deformation in three-point bending testing.

**Figure 4 materials-16-01909-f004:**
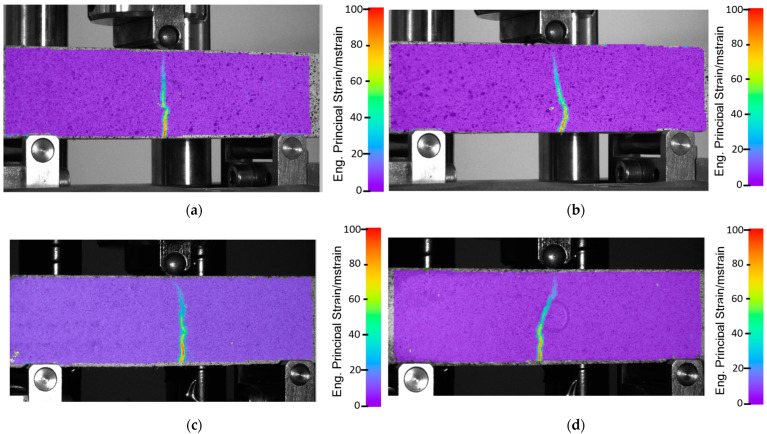
Results of the DIC analyses during flexural strength tests of CGCB samples in (**a**) horizontal, and (**b**) vertical orientation; and CGCB with scaffolding in (**c**) horizontal, and (**d**) vertical orientation.

**Figure 5 materials-16-01909-f005:**
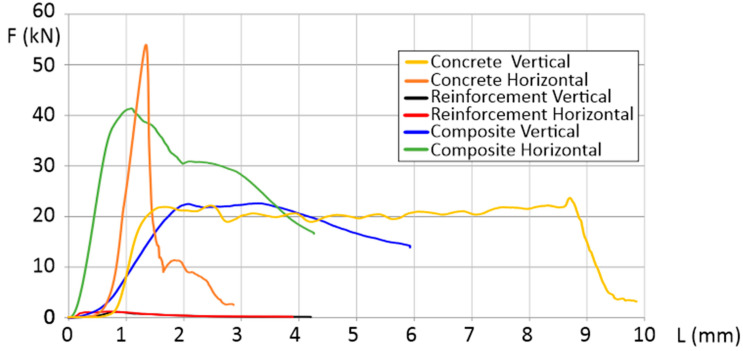
Force as a function of deformation in compression testing.

**Figure 6 materials-16-01909-f006:**
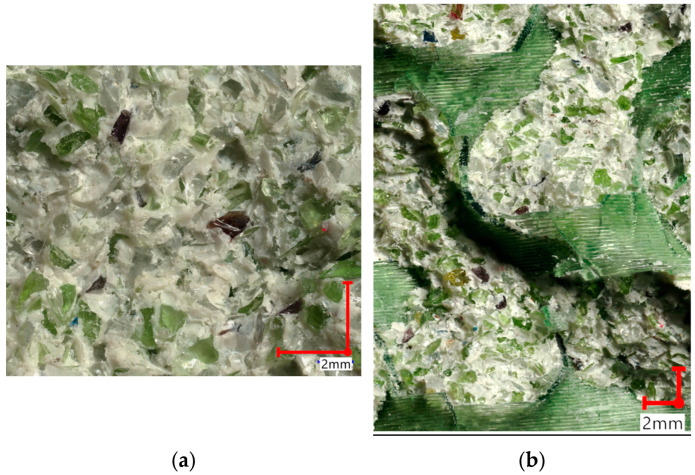
The fracture surface of the tested samples: (**a**) without scaffolding (**b**) with internal, polymeric scaffolding.

**Figure 7 materials-16-01909-f007:**
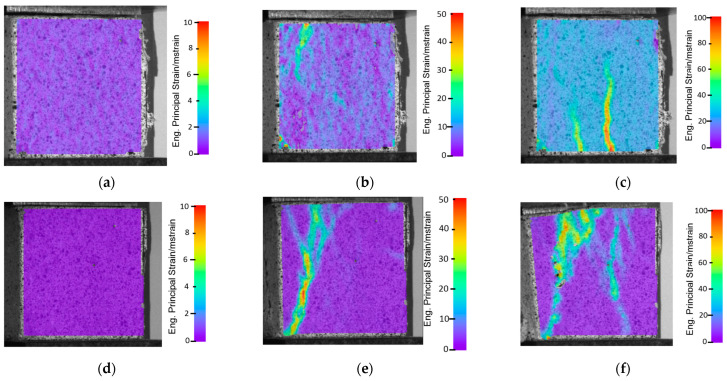
DIC images of compressive strength tests of samples without scaffolding (**a**–**f**), and with scaffolding (**g**–**l**): in vertical arrangement (**a**) at the start, (**b**) after 1 mm of strain, (**c**) before sample breaking; and in horizontal arrangement (**d**) at the start, (**e**) after 1 mm of strain, (**f**) before sample breaking; and samples with scaffolding in vertical arrangement (**g**) at the start, (**h**) after 1 mm of strain, (**i**) before sample breaking; and in horizontal arrangement (**j**) at the start, (**k**) after 1 mm of strain, (**l**) before sample breaking.

**Table 1 materials-16-01909-t001:** Composition of the aggregate and the binder [25].

Compositions		SiO_2_	Al_2_O_3_	Fe_2_O_3_	CaO	MgO	SO_3_	Na_2_O	K_2_O	TiO_2_	Cl
Unit (vol.%)	Cement	19.5	4.9	2.9	63.3	1.3	2.8	0.1	0.9	-	0.05
	Glass	70.0–74.0	0.5–2.0	0.0–0.1	7.0–11.0	3.0–5.0	-	6.0–8.0	7.0–9.0	0.0–0.1	-

**Table 2 materials-16-01909-t002:** Properties of the aggregate and the binder [25].

Property	Specific Surface Area [m^2^/kg]	Specific Gravity [kg/m^3^]	Initial Setting Time [min]	Average Compressive Strength After 28 days [MPa]
Cement	437	3090–3190	176	68.2
Glass	100	2450	-	-

**Table 3 materials-16-01909-t003:** CGM mix proportion (1 m^3^).

Mix symbol	Cement [kg]	Water [kg]	Chemical Admixture [kg]	Waste Glass Powder [kg]	Waste Glass Aggregate [kg]
CGM	480	140	4.8	117.8	1782.2

**Table 4 materials-16-01909-t004:** Thermal properties of composite materials.

Sample Symbol	Thermal Conductivity [W/mK]	Thermal Diffusivity (µm^2^/s)	Specific Heat (MJ/m^3^K)
CGM [14]	0.99 ± 0.05	0.61 ± 0.03	1.64 ± 0.01
CGCB in this study	0.87 ± 0.05	0.69 ± 0.03	1.31 ± 0.01
CGCB by Małek et al. [14]	0.91 ± 0.05	0.66 ± 0.03	1.48 ± 0.01

## Data Availability

Data available on request.

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
