# Peer review of "Performance Properties of Cement–Glass Composite Bricks (CGCB) with Additively Manufactured (AM) Polymeric Scaffolding"

_materials, 2023, doi:10.3390/ma16051909_

Round 1
Reviewer 1 Report
The topic of the paper is interesting but the paper has not been structured properly. The novelty of the paper is missing. The following are required to be considered by the authors before the paper being accepted.
·
· 1) The gap in the literature has not been stated and the novelty of the current research is missing. Therefore, these must be addressed in the revised version.
· 2) The current conclusions needs to be amended as it is not presented properly. Therefore, authors are firstly required to state the main aim of the current research at the conclusion and then listing the main findings.
Author Response
Dear Reviewer,
On behalf of all authors, I would like to thank you for taking the time to read our manuscript and put in your comments which allowed us to improve the quality of our work. Below you can find our answers related to each of your comments. Each correction based on your comments is yellow-highlighted.
- The gap in the literature has not been stated and the novelty of the current research is missing. Therefore, these must be addressed in the revised version.
Ad.1. Thank you very much for this comment. We added additional sentences which highlighted the research gap and novelty of our research. We formed the following sentences:
Despite significant growth in pro-ecological movements in building engineering, there are a lot of issues that have to be described from the material’s behavior point of view. An additional factor in the case of this research is also usage of polymeric scaffolding which needs to be motivated. In the case of previous research [14], there was a positive effect of the use of scaffolding, but the reinforcing structure did not properly penetrate the scaffold. Therefore, in this study, the AM polymeric scaffolding has been used for the production of CGCB brick with the use of waste glass as an aggregate material. The geometry of the scaffold has been modified to allow proper penetration of the waste glass parts. In the available research, there is not enough data about using a mixture of waste polymers (as scaffolding) and waste glass (as reinforcing structures). Hence, mechanical and thermal analyses of such material were made to determine the basic performance properties of such kinds of composite materials. A suggested approach could be useful in further analysis of using waste materials in building engineering.
- The current conclusions needs to be amended as it is not presented properly. Therefore, authors are firstly required to state the main aim of the current research at the conclusion and then listing the main findings.
Ad.2. At the beginning of the conclusion part we attached a few sentences about the aim of our research:
The main aim of this research work was to determine the mechanical and thermal behavior of the proposed CGCB bricks. The additional aim was based on the comparison between scaffolded, and non-scaffolded test samples. Such kind of approach allowed for deep analysis of using additional polymeric structures. Obtained results The research results revealed positive physical properties (thermal and mechanical) of a new type of concrete with glass particles, additionally reinforced by the additively manufactured polymeric structure.
Reviewer 2 Report
This paper contains the research results on the flexural behavior of cement-glass composites reinforced with polymer scaffolding with special geometry.
(main comment)
- This study only explained the behavior for the bending test. Of course, applying an effective measurement method such as DIC analysis is an excellent approach. However, there is no analytical approach to the physical behavior. In other words, it is more like a test report than a scientific paper. Therefore, in order for this thesis to be published as a scientific paper, analysis on mechanical behavior is required.
- There are no variables in this experiment. It is necessary to review the various geometries of scaffolding and the degree of filling rate of cement mixture in scaffolding.
(minor comments)
- Fig. 1 is not required
- It is common to express the mix proportion of concrete as "unit weight per volume (kg/m3)"
Author Response
Dear Reviewer,
On behalf of all authors, I would like to thank you for taking the time to read our manuscript and put in your comments which allowed us to improve the quality of our work. Below you can find our answers related to each of your comments. Each correction based on your comments is green-highlighted.
- This study only explained the behavior for the bending test. Of course, applying an effective measurement method such as DIC analysis is an excellent approach. However, there is no analytical approach to the physical behavior. In other words, it is more like a test report than a scientific paper. Therefore, in order for this thesis to be published as a scientific paper, analysis on mechanical behavior is required.
Ad.1. Thank you very much for this opinion. In the case of our research, there are too many unknown factors that determine the material’s behavior. Only the case of the usage of additively manufactured scaffolding requires complicated FEA (Finite Elements Analysis). In the case of such a complex composite, a reliable analysis are unfortunately beyond our research reach. We agree with the statement, that in the present form, the DIC results look more like a report. That is why we made a discussion about the available research works. You can find it green-highlighted in our manuscript, and also below this answer:
- Bending tests:
Such kind of obtained DIC results does not significantly differ from the available literature data. Shah et al. [31] analyzed mortar bricks with the use of the DIC method during three-point bending tests. The authors registered a single crack tip along the sample part with significant amounts of microcracks in the fracture zones that form ahead of the traction-free crack which is a property of quasi-brittle material. A visible increase in total deformation of horizontally oriented composite samples (Fig.3) decreases the typical behavior of cracking (which is proved by a more complex crack tip in the case of horizontally oriented samples). Quite a different approach, more suitable for CGCB samples with AM scaffolding was suggested by Wu et al [32], where they used a cohesive crack concept that allows describing the strain-softening behavior of the material. Such kind of approach allows for finding a relationship between the fracture process zone length and the crack length. Obtained results with the use of the DIC method help to better study the behavior of the crack extension resistance curve of the material. Based on this method and DIC measurements, it could be observed that the length increases during crack propagation but decreases after the fracture process zone is fully developed at a crack extension.
- Compressive tests:
A useful method that could be used for a proper description of that kind of material’s behavior was described by Zhou et al. [34] for rock-like specimens. The stress-strain curves in this methodology are divided into two stages:
- the strain-softening stage;
- the residual strength stage.
The strain-softening stage is characterized by rapid crack evolution and further propagation which leads to a rapid drop in the axial stress. With the increase of uniaxial compressive loads, the axial stresses decrease to the residual strengths in the materials with different brittleness indices (i.e. polymeric scaffold or waste glass reinforcement). Such kind of approach would be possible by the analysis of particular cells of the scaffold.
- There are no variables in this experiment. It is necessary to review the various geometries of scaffolding and the degree of filling rate of cement mixture in scaffolding.
Ad.2. We decided not to include additional factors, because such a move would generate a lot of additional results that would blur the comparison between those two most important cases (with and without scaffolding). Please note that using waste glass is a kind of variable – an additional scaffold is a further step. Based on the previous research [M. Małek, K. Grzelak, Ł. Waldemar, M. Jackowski, J. Kluczyński, I. Szachogłuchowicz, J. Torzewski, J. Łuszczek, Cement-glass composite bricks ( CGCB ) with interior 3D printed PET-G scaffolding, 52 (2022). doi:10.1016/j.jobe.2022.104429] we designed a new type of structure that enabled better distribution of the waste glass. We tried to focus on the exact structure to prepare a kind of deeper research analysis instead of the case report. That is why we resigned on the preparation of the wide review of different combinations.
- - Fig. 1 is not required
Ad.3. It has been removed
- It is common to express the mix proportion of concrete as "unit weight per volume (kg/m3)"
Ad.4. We put an additional table 3 which shows the proper mix portion.
Reviewer 3 Report
The following points need to be addressed before acceptance.
1) Line 85-101. Write only the outcome of the reviewed literature. Unwanted maybe delete
2) Why do authors use Material Extrusion (ME) technology?
3) Is there any delamination or warping issue addressed during printing?
4) What about the moisture effect during FFF or FDM printing?
5) Line 180 - What is Ten 150?
6) Line 202 - "The analysis of the densities of the finished brick and the cured mortar revealed small variations in the values". Where are the variation values?
7) Line 205-208 - The sample's density was reduced by about 5% (a difference of 106 kg/m3) after 3D-printed scaffolding was 206 added to the brick matrix. This is because plastic weighs a lot less than cement-glass mortar, which was removed in part to make room for the scaffolding from a 3D printer in the brick matrix." The statement is confusing. Rewrite and provide an appropriate statement.
8) Line 211 - What is the 3Dhon structure of scaffolding?
9) Why is there a need to investigate the Thermal tests for CGM and CGCB?
Author Response
Dear Reviewer,
On behalf of all authors, I would like to thank you for taking the time to read our manuscript and put in your comments which allowed us to improve the quality of our work. Below you can find our answers related to each of your comments. Each correction based on your comments is blue-highlighted. Additionally, based on your comment. We asked for an additional language check. Please see the attached pdf file with marked changes.
- Line 85-101. Write only the outcome of the reviewed literature. Unwanted maybe delete.
Ad.1. We made proper corrections based on your advice. Please note that the yellow-highlighted part was made according to the other reviewer’s comment.
- Why do authors use Material Extrusion (ME) technology?
Ad.2. We use such kind of technology because it is characterized by the highest availability of recycled materials (PET and PET-G). Also, machines which are using such technology are inexpensive and characterized by high availability.
- Is there any delamination or warping issue addressed during printing?
Ad.3. We did not struggle with such phenomena because we used PET-G material, characterized by one of the most significant adhesions (between layers and to the substrate plate). Additionally, we used process parameters recommended by the supplier. Based on visual analysis and strength tests of the polymeric scaffold, we were assured of the proper manufacturing process.
- What about the moisture effect during FFF or FDM printing?
Ad.4. PET-G is a hydrophobic material. Before the process, it was dried in the following conditions. We attached this information also in the manuscript: „Before the process, the material was dried in the laboratory dryer for 4 hours at 65°C (using the manufacturer's instructions).”
- Line 180 - What is Ten 150?
Ad. 5. It means that we tested 10 specimens. To clarify we rephrased the sentence: Ten specimens in a form of cubes 150 mm x 150 mm x 150 mm cubes in total were evaluated to determine all of the CGM’s thermal characteristics of CGM.
- Line 202 - "The analysis of the densities of the finished brick and the cured mortar revealed small variations in the values". Where are the variation values?
Ad.6. We decided to remove this sentence. Thanks to your comment we saw that this phrase may be misleading.
- Line 205-208 - The sample's density was reduced by about 5% (a difference of 106 kg/m3) after 3D-printed scaffolding was 206 added to the brick matrix. This is because plastic weighs a lot less than cement-glass mortar, which was removed in part to make room for the scaffolding from a 3D printer in the brick matrix." The statement is confusing. Rewrite and provide an appropriate statement.
Ad.7. We rephrased this part: Using additional polymeric scaffolding caused a The sample's density was reduced re-duction by about 5% (a difference of 106 kg/m3) after 3D-printed scaffolding was added to the brick matrix. Such a phenomenon was caused by a significant mass difference between cement-glass mortar and PET-G material.
- Line 211 - What is the 3Dhon structure of scaffolding?
Ad.8. We rephrased the sentence. Małek et al. [14] presented similar trends for the use of three-dimensional structure“3Dhon” of scaffolding
- Why is there a need to investigate the Thermal tests for CGM and CGCB?
Ad. 9. Put additional information about this issue in the introduction part. It has been blue-highlighted.
„Due to the difference in the materials used for the production of the concrete composite, i.e. the ceramic material and the polymer, the authors noticed that there is a physically justified difference between the thermal conductivity of both materials. As a result of their physical connection, there are no reaction processes between the concrete and the polymer, therefore chemical adhesion does not occur in this case. According to the literature, in such a case, the combination of polymers and ceramic materials leads to the so-called mechanical adhesion, i.e. due to the geometry of the 3D print, the structure was "anchored" in the designed concrete. In that case, there is a need to check whether, apart from the increase in mechanical parameters, there is no decrease in other parameters that are a determinant for concrete in construction, i.e. its insulation. According to the results of the obtained tests, the insulating power of such a composite has increased, which reassured the authors of the validity of their work. The increase in insulation results was achieved by creating an air layer between the contact boundaries of both materials, which is an ideal insulator.”
Round 2
Reviewer 1 Report
I am happy with the corrections and the paper is ready to be published.
Reviewer 3 Report
Dear Authors,
The paper is accepted in its present form.